# SteerLM: Attribute Conditioned SFT as an (User-Steerable) Alternative to RLHF

**Yi Dong, Zhilin Wang, Makesh Narsimhan Sreedhar, Xianchao Wu, Oleksii Kuchaiev**

NVIDIA

{yidong, zhilinw, makeshn, xianchaow, okuchaiev}@nvidia.com

## Abstract

Model alignment with human preferences is an essential step in making Large Language Models (LLMs) helpful and consistent with human values. It typically consists of supervised fine-tuning (SFT) and reinforcement learning from human feedback (RLHF) stages. However, RLHF faces inherent limitations stemming from a complex training setup and its tendency to align the model with implicit values that end users cannot control at run-time. Moreover, reward models in RLHF stage commonly rely on single-dimensional feedback as opposed to explicit, multifaceted signals that indicate attributes such as helpfulness, humor, and toxicity. To address these limitations, we propose STEERLM, a supervised fine-tuning method that empowers end-users to control responses during inference. STEERLM conditions responses to conform to an explicitly defined multi-dimensional set of attributes, thereby empowering a steerable AI capable of generating helpful and high-quality responses while maintaining customizability. Experiments show that STEERLM trained on open source datasets generates responses that are preferred by human and automatic evaluators to many state-of-the-art baselines trained with RLHF while being much easier to train. Try STEERLM at https://huggingface.co/nvidia/SteerLM-llama2-13B

## 1 Introduction

Training LLMs on extensive text corpora has demonstrated remarkable capabilities, leading to state-of-the-art performance on numerous tasks (Brown et al., 2020; Kaplan et al., 2020). However, this does not automatically make language models effective in responding to user instructions (Wei et al., 2022; Sanh et al., 2022). To better align LLMs to human preferences, the most effective approach has been to perform SFT followed by the application of RLHF (Wang et al., 2023a; Chiang et al., 2023; Peng et al., 2023). In SFT, human

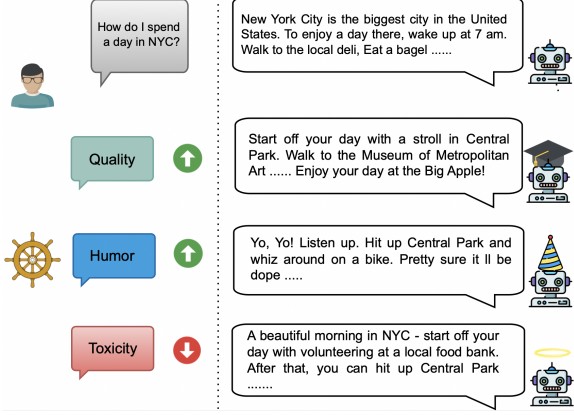

Figure 1: STEERLM can be used to improve the quality of language model responses, similar to RLHF. Additionally, STEERLM allows users to define additional attributes such as humor and (low) toxicity at inference time, to steer model responses.

annotators provide demonstrations of instructions and responses for the model to imitate (Taori et al., 2023; Zhang et al., 2023). RLHF goes a step further to enable models to generate responses that human annotators prefer to alternative responses (Bai et al., 2022; Ouyang et al., 2022; Köpf et al., 2023a).

However, despite its success, there are limitations to this approach. First, using SFT alone does not allow the model to distinguish between high-quality and low-quality responses leading to lower performance than RLHF (Wang et al., 2023a). Using RLHF for model alignment however, substantially increase the complexity of the training setup (Snell et al., 2023; Yuan et al., 2023), limiting its public adoption (Zhang et al., 2023; Dettmers et al., 2023; Zhou et al., 2023). Furthermore, RLHF treats human preference of model responses as mono-dimensional without regard for the diversity of aspects (e.g. helpfulness, humor, toxicity) that contribute to such preferences (Bai et al., 2022; Ouyang et al., 2022) and thereby limiting users' ability to adjust individual aspects at inference time

based on their use cases.

To address these limitations, we introduce STEERLM, a novel approach to model alignment through SFT that overcomes the limitations associated with conventional SFT and RLHF methods. Similar to RLHF, STEERLM incorporates additional reward signals by leveraging annotated attributes (e.g., quality, humor, toxicity) present in the Open-Assistant (Köpf et al., 2023a) dataset for each response. To emulate the process of rating responses, an Attribute Prediction model (§3.1) is trained and employed to annotate datasets (§3.2) containing dialogues that can be decomposed into prompt-response pairs. By utilizing this combined diverse dataset comprising prompts, responses, and predicted attributes, we train the generation of responses to be conditioned (§3.3) on both the prompt instructions and the annotated attributes, enabling STEERLM to effectively capture human preferences and generate responses that align with them.

STEERLM also exhibits enhanced versatility compared to RLHF, allowing for the flexible utilization of various attributes during inference. To further enhance the adherence of STEERLM to the specified attributes during inference, we introduce an additional training recipe (§3.4) which includes augmenting the training data with denoised, diverse, and high-quality examples. We open-source code for STEERLM on NVIDIA NeMo toolkit (Kuchaiev et al., 2019). In summary, our key contributions are:

- We introduce STEERLM as a simple alternative for language model alignment that utilizes only the language modeling objective,

- We demonstrate the efficacy of STEERLM 43B on the Vicuna benchmark where it outperforms state-of-the-art baselines including RLHF models such as *ChatGPT-3.5*,

- We highlight the flexible and customizable nature of STEERLM 43B where users can customize attributes at inference-time facilitating a wide variety of applications.

## 2 Related Work

**Model alignment using SFT** Fine-tuning language models on multiple tasks enable them to follow many types of instructions and perform tasks outside of those they were trained on (Sanh et al., 2022; Wei et al., 2022). However, language models typically generate short and robotic responses when supervised finetuned on academic data sets. On the other hand, models can generate high quality human-like response when trained with high-quality human demonstrations (Conover et al., 2023; Ouyang et al., 2022). Taori et al. (2023) shows using data generated by OpenAI's text-davinci-003 model, can train a model in a cost-effective manner.

Using only SFT for model alignment became popular recently because of the ease of its training setup. Zhang et al. (2023) and Peng et al. (2023) trained models using SFT based on responses generated by OpenAI models while Dettmers et al. (2023) and Köpf et al. (2023b) used the crowd-sourced Open Assistant Dataset and Zhou et al. (2023) used a small proprietary data set. Wang et al. (2023b) and Taori et al. (2023) trained models using bootstrapped datasets from the language model itself. Luo et al. (2023) showed that language model can learn to solve complex instructions by evolving on the complexity and breath of instructions. Wang et al. (2023a) compared many open-source data sets used to perform instruction tuning using SFT but found them to under-perform commercial models trained using RLHF.

**Model alignment using RLHF** Building on foundational work on RL in games and robotic simulations (Christiano et al., 2017; Schulman et al., 2017), many have had success applying RLHF to improve the instruction following ability of LLM by giving a reward proportional to the relative desirability of the response (Ouyang et al., 2022; Bai et al., 2022). Such an approach has been shown to benefit downstream tasks like question-answering (Nakano et al., 2022) and summarization (Stiennon et al., 2022). However, the complexity of the training setup (Rafailov et al., 2023; Snell et al., 2023) remains a hurdle in the widespread adoption of RLHF. Many have attempted to overcome this by migrating RLHF training to an offline setting (Snell et al., 2023), casting the problem as conditional sequence modeling (Chen et al., 2021), directly optimizing LMs with labeled preference data (Rafailov et al., 2023), or ranking responses to align the models (Dong et al., 2023; Yuan et al., 2023), but limited progress has been made (Wang et al., 2023a).

Another limitation unaddressed by related works lies in the use of a single-dimensional reward function for evaluating human preferences of model

responses since human preferences are based on a multitude of real-world objectives (e.g. helpfulness, humor, toxicity), which also vary across domains (Nadal and Chatterjee, 2019; Lopez-Paz et al., 2022). Given such multi-dimensional attributes, current approaches (Bai et al., 2022; Ouyang et al., 2022; Dong et al., 2023; Yuan et al., 2023) are only capable of generating responses with high reward scores, despite low reward scores on various attributes being relevant in certain situations, such as modelling realistic gaming NPCs who are capable of generating high toxicity responses.

**Attribute Grounded Generation** Many researchers have explored grounding text with various attributes in Dialogue tasks. Rashkin et al. (2019) modelled chit-chat conversations grounded in emotions such as 'Angry', 'Embarrassed' or 'Joyful' while Smith et al. (2020) conditioned chit-chat dialogue with conversational styles such as 'Curious', 'Sympathetic' or 'Knowledgeable' . Zhang et al. (2018) and Wang et al. (2022) conditioned dialogues based on personal attributes such as their hobbies. Meta et al. (2022) conditioned dialogues in the game of Diplomacy using the expected player skill. However, such grounding has only been explored in narrow-defined tasks with a single attribute. Our approach seeks to condition the generation of responses in general open-domain conversations covering tasks like code assistance, writing poems and planning tasks, using multiple attributes (e.g., quality, humor and toxicity).

## 3 SteerLM

We propose STEERLM, a simple and novel approach to align language models to follow user instructions. Trained solely using the language modeling objective, it offers a computationally efficient alternative to other techniques like RLHF. Specifically, STEERLM comprises 4 steps as illustrated in Fig. 2.

### 3.1 Step 1. Attribute Prediction Model

Similar to the reward model in RLHF, the Attribute Prediction Model in STEERLM is designed to predict human preference of model responses to improve model alignment. Compared to a monolithic reward signal in RLHF, the attribute prediction model can be used to predict various attributes that are considered to be important in generating good responses (high quality, low toxicity, and varying humor levels depending on context).

We use the Open Assistant (OASST) dataset $D$, where each sample contains a prompt $x$, a response $y$ as well as a set of attributes $v$. To model these attributes, we first scale each attribute (originally a float between 0 and 1) into an integer between 0 and 9 and then obtain a linearized representation of the value attributes $v$. The attributes we select look like `quality:6,toxicity:0,humor:9,creativity:0, violence:0,helpfulness:5,not_appropriate:0`.

$$\mathcal{L}_{APM} = -\mathbb{E}_{(x,v,y)\sim D} \sum_t \log P_\theta(v_t|x,y,v_{<t})$$
(1)

Conditioning on $x$ and $y$, $v$ is the target output for the language model as expressed in Eq. 1.

### 3.2 Step 2. Annotating Datasets using Attribute Prediction Model

Compared to using human-annotated attributes directly, training an Attribute Prediction Model can allow other datasets (e.g. HH-RLHF dataset) to be annotated. This helps improve the diversity of training data which is important for Step 3 Attribute Conditioned SFT. Moreover, it has been observed that crowdsourced human-annotated data often suffers from noise, arising from factors such as misinterpretation of instructions, inadequate expertise/education in annotating responses, and limited proficiency in language comprehension (Köpf et al., 2023a). Furthermore, there exists a lack of calibration among annotators, with some individuals applying more stringent criteria when assigning full scores (Bai et al., 2022; Ouyang et al., 2022). By employing an Attribute Prediction Model, it becomes possible to mitigate these issues by denoising the human-annotated attributes and calibrating scores across annotators.

$$\arg\max_{v_t} P_\theta(v_t|x,y,v_{<t}) \qquad (2)$$

We annotate samples by greedily decoding the value attributes for pairs of prompts and responses using the Attribute Prediction Model(as shown in Eq. 2), in order to construct the attribute annotated dataset $D'$.

### 3.3 Step 3. Attribute Conditioned SFT

Attribute-conditioned SFT is an extension of regular SFT that enables incorporating reward signal information through attribute labels. This allows learning from both high and low quality responses in a manner similar to the established SFT+RLHF pipeline (Bai et al., 2022; Ouyang et al., 2022).

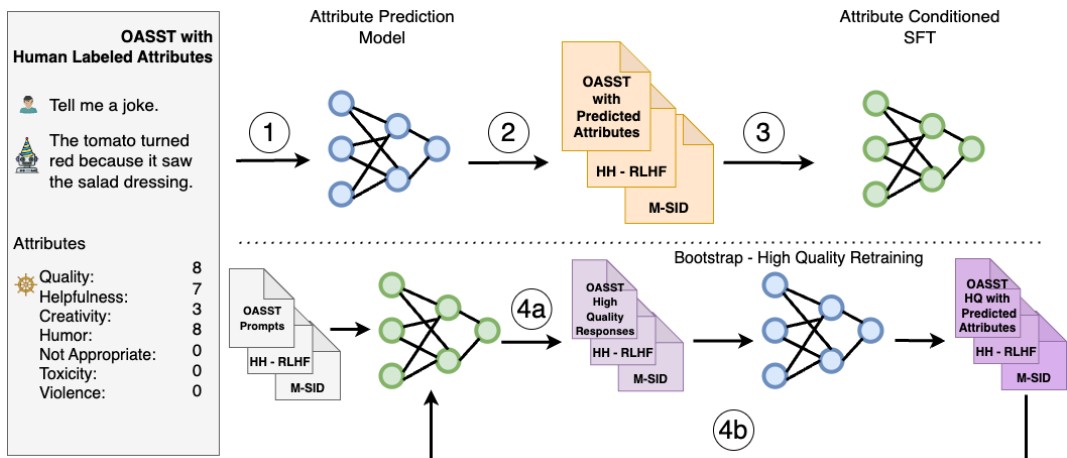

Figure 2: **STEERLM**: **Step 1.** The base language model is trained to assess the quality of responses by predicting attribute values. **Step 2.** The attribute prediction model is used to annotate response quality across diverse datasets. **Step 3.** Given a prompt and desired attribute values, a new base model is fine-tuned to generate responses that align with the specified attributes. **Step 4.** Multiple responses are sampled from the fine-tuned model in Step 3, specifying maximum quality. The sampled responses are evaluated by the trained attribute prediction model, leading to another round of fine-tuning.

Attribute-conditioned SFT only requires an offline annotated dataset, as created in Step 2, rather than online sampling and evaluation of responses like in RLHF. By utilizing a purely offline training approach, this greatly simplifies the training configuration compared to RLHF's heterogenous setup. In particular, it avoids the complexity of online data generation/evaluation and eliminates the training slowness resulting from memory bandwidth-constrained online inference in RLHF. Using the attribute-annotated train datasets $D'$ from the Step 2, we train a model to generate a response $y$, conditioning on the value attributes $v$ and the prompt $x$. The loss function is:

$$\mathcal{L}_{ACSFT_1} = -\mathbb{E}_{(x,v,y)\sim D'} \sum_t \log P_\phi(y_t|v, x, y_{<t})$$

### 3.4 Step 4. Bootstrapping with High Quality Samples

By sampling the policy network, RLHF effectively navigates the response space of language models and identifies responses that are of various qualities. The response samples are subsequently utilized to influence and shape the behavior of language models according to their reward values. In Step 4 of STEERLM, the objective is to accomplish a similar objective by leveraging the Attribute Conditioned SFT and Attribute Prediction models from the previous steps.

**Step 4a**  To ensure that we obtain a diverse set of responses, we first enumerate a set of all possible

attribute value combinations from the annotated datasets used for training. By filtering the combinations to explicitly have the value for *quality* to be 9 (i.e. highest possible value), we get a subset of attribute strings representing a high quality set $V$. We uniformly sample from this high quality set $V$ to get the attribute string $v'$. By combining $v'$ with prompts from the same training datasets, we use the top-$k$ (=50) sampling to generate multiple responses using the Attribute Conditioned Supervised Fine-Tuning approach as shown in Eq. 3.

$$\arg\_topk_{y'_t} P_\phi(y'_t|x, v', y'_{<t}) \tag{3}$$

This allows us to obtain a wide variety of responses for each prompt and increases the diversity of the sampled dataset $D'' = \{(x, y')\}$.

The Attribute Prediction Model (§3.1) with greedy sampling is used to evaluate the generated responses $y'$ giving predicted attribute values $v''$:

$$\arg\max_{v''_t} P_\theta(v''_t|x, y', v''_{<t}) \tag{4}$$

This gives us the dataset $D''' = \{(x, y', v'')\}$ where each tuple consists of a prompt from the original training datasets, a sampled response and its corresponding predicted attribute values.

**Step 4b**  We use the sampled responses and the corresponding predicted attributes from $D'''$ to perform the second round of Attribute-conditioned

SFT, effectively allowing us to bootstrap the training of the model on its own responses:

$$\mathcal{L}_{ACSFT_2} = -\mathbb{E}_{(x,v'',y')\sim D'''} \sum_t \log P_\phi(y'_t|v'',x,y'_{<t})$$

## 4 Experiments

In this section, we elaborate on our choice of training datasets, base model, implementation details, and evaluation setup.

### 4.1 Training Datasets

We use the following open-source, commercially-available instruction-tuning datasets. In addition, we explain how collecting new data for SteerLM can be less costly than doing so for RLHF in Appendix A.3.

**OASST** Open Assistant dataset (Köpf et al., 2023a) was used to train an Attribute Prediction Model, as well as to perform Attribute Condition SFT. This dataset contains 13 human-labeled attributes for each response with a score ranging from 0 to 1. We choose 7 of them that are most relevant for guiding the language model to align with human preferences: quality, toxicity, violence, helpfulness, creativity, humor and inappropriateness. Other attributes such as hate_speech, lang_mismatch and pii (personal identifiable information) are not useful to steer at inference time since these are attributes that we always want to keep as False. In order to leverage STEERLM's ability to learn from both positive and negative responses, we use all the released 48.8K conversations for training and do not filter the dataset.

**HH-RLHF** The Helpful and Harmless - Reinforcement Learning from Human Feedback dataset (Bai et al., 2022) does not provide human labeled attribute values. In order to improve the diversity of prompts and responses, we utilize the trained Attribute Prediction model to annotate the responses. For our Attribute Conditioned SFT process, we utilize all 348.6K conversations from the dataset.

**M-SID** Model Self-Identification Dataset (Chiang et al., 2023) is a small set of 910 prompt-response pairs[1] used to answer questions relating to identity such as "Who are you?" and "Who created you?". This dataset is also included as part of Attribute Conditioned SFT training.

[1]Adapted from https://github.com/lm-sys/FastChat/blob/v0.2.1/playground/data/dummy.json

### 4.2 Base Models for STEERLM

**STEERLM 43B** The 43B base language model employed in this study has been trained on a diverse corpus encompassing various multilingual data sources, including web crawl, news articles, books, scientific publications from arXiv, and code repositories. Having been trained with 1.1 trillion tokens, it is comparable to LLaMA's 30B and 65B models, which were trained on 1.4 trillion tokens. This base model is designed for general-purpose language understanding tasks and does not have any domain-specific fine-tuning. We utilize this base model as the backbone for both Attribute Prediction and Attribute Conditioned SFT.

**STEERLM 13B** We also apply the SteerLM methodology on a popular, widely-available model: Llama 2 13B base model (Touvron et al., 2023).

### 4.3 Training details

The training of both the Attribute Prediction Model and Attribute Conditioned Supervised Fine-Tuning model was conducted utilizing a cluster comprising 16 A100-DGX nodes, each equipped with 8 A100-80GB GPUs. The training process involved a global batch size of 128, spanning 5 epochs, with a maximum sequence length of 4096 tokens. The Adam optimizer was utilized with a learning rate of 5e-6 and a weight decay of 1e-2. The selection of the optimal Attribute Prediction model checkpoint was determined based on the lowest loss observed on the validation set, while the optimal checkpoint for the Attribute Conditioned SFT model was selected based on the highest validation quality observed on holdout validation sets. Detailed templates for Attribute Prediction Model and Attribute Conditioned SFT are found in Appendix §A.1 and §A.2.

### 4.4 Evaluation

**Baseline Models** We compare our approach against several state-of-the-art instruction-following models. These baselines include OpenAI ChatGPT 3.5, OpenAI text-davinci-003, Guanaco 65B (Dettmers et al., 2023), Vicuna 13B (Chiang et al., 2023), and OASST LLaMA 30B RLHF (Köpf et al., 2023a). Furthermore, to showcase the differentiation between RLHF and SFT, we also include OASST LLaMA 30B SFT (Köpf et al., 2023b), which solely employs SFT instead of RLHF for alignment purposes.

**Response Generation**  In accordance with the methodologies described in Peng et al. (2023) and Dettmers et al. (2023), we employ the GPT-4 model to conduct an evaluation of our proposed approach using the Vicuna benchmark (Chiang et al., 2023). This benchmark comprises a diverse set of 80 single-turn prompts encompassing various domains, including knowledge-intensive question answering, historical counter-factuals, long-form document writing, commonsense questions, role-plays, fermi problems, general knowledge questions, coding, and math problems. Each model is tasked with generating a response with a maximum sequence length of 1024, utilizing default hyperparameters. For all the STEERLM models and OASST LLaMA 30B models, the decoding strategy employed is greedy decoding. Additionally, for STEERLM model responses, the attributes "quality" and "helpfulness" are fixed at a value of 9, while all other attributes are set to 0. In the case of Guanaco 65B, generations are obtained using top $p = 0.9$ sampling with a temperature of 0.7, and Vicuna 13B uses a temperature of 0.7[2].

**Automatic Evaluation**  In evaluating each prompt, the prompt alongside the responses made by the two competing models are given to GPT-4 (Chiang et al., 2023; Dettmers et al., 2023). GPT-4 is then required to give a score between 1 and 10 for each response. While we only report scores for comparing each model against ChatGPT 3.5, we compare every pair of models against each other for subsequent use in ELO rating calculations. The cumulative score obtained over the 80 questions for each model is then compared to the cumulative score achieved by ChatGPT 3.5. This facilitates the assessment of each model's performance as a percentage of ChatGPT 3.5's performance. It is important to note that the ordering of the responses from the two models during evaluations can influence the evaluation results, as previously observed by Dettmers et al. (2023). To mitigate this potential bias, we calculate the mean performance of each model in both possible response orderings.

| Model Name | ChatGPT 3.5 Score | Model Score | % of ChatGPT 3.5 |
|---|---|---|---|
| STEERLM 43B | 629.25 | 655.75 | **104.2** |
| STEERLM 13B | 617.75 | 634 | 102.6 |
| Guanaco 65B | 631.25 | 646.25 | 102.4 |
| ChatGPT 3.5 | - | - | 100.0 |
| Vicuna 13B | 641.75 | 636.75 | 99.2 |
| LLaMA 30B RLHF | 650.25 | 612.75 | 94.2 |
| LLaMA 30B SFT | 631.5 | 610 | 93.2 |
| text-davinci-003 | 665.25 | 599.5 | 90.1 |

Table 1: Automatic evaluation. Evaluations are highly consistent with +/- 0.2% difference across evaluation runs on identical model generations.

**Human Evaluation**  To minimize the risk of annotator fatigue and the potential for hasty evaluations, we partitioned the 80 prompts within the Vicuna Benchmark into four distinct groups, with each group consisting of 20 prompts. This partitioning strategy helps ensure that the human annotators (12 in total) are able to provide thorough evaluations without skimming through the responses, as it can be demanding to compare multiple responses simultaneously. Volunteer human annotators were specifically chosen for this task instead of utilizing crowd workers, such as Amazon Mechanical Turkers, as it enables us to carefully control for annotators with a university education and coding background, which are crucial for effectively evaluating these model responses. Consequently, this approach resulted in a higher degree of consistency among our annotators (Fleiss' $\kappa = 0.46$) in comparison to similar studies that employed crowd workers (Dettmers et al., 2023).

However, this choice limited the number of models that we could compare, and we selected the three best-performing models from automatic evaluations for the human evaluation (excluding STEERLM 13B given its similarities to STEERLM 43B). The prompt along with the responses from the different models were shown to the annotators, and they were asked to rank them in order of preference (using the UI in Appendix §A.5). During the annotation process, the human annotators remained unaware of the specific model used for each response, and to prevent any potential bias, the order of the responses was randomly shuffled. Annotations were carried out utilizing an A/B forced choice approach, requiring the annotators to subjectively determine their preferred response between the given options.

---

[2]Guanaco 65B generations at `https://github.com/artidoro/qlora/blob/main/eval/generations/vicuna/65b-guanaco-vicuna-generations-topp0.9-temp0.7.jsonl` and Vicuna 13B generations at `https://github.com/lm-sys/FastChat/blob/main/fastchat/eval/table/answer/answer_vicuna-13b.jsonl`

**Elo rating** Using both automatic and human pair-wise model comparisons, we calculate an Elo rating for each model to show the performance relative to all other models. For each prompt, scores are first converted into a tie, win or loss between two models based on their scores. Following Chiang et al. (2023) and Dettmers et al. (2023), we start with a score of 1000 and $K = 32$ and repeat the procedure 10000 times to account for the order in which model pair comparisons are used to calculate their ELO rating. Additionally, we also present the expected win rate against ChatGPT 3.5, for better interpretability.

### 4.5 Results

| Model Name | Elo Rating | 95% CI | Win Rate (%) vs ChatGPT 3.5 |
|---|---|---|---|
| Automatic Evaluation | | | |
| STEERLM 43B | **1139** | 1048-1224 | **66** |
| STEERLM 13B | 1110 | 1022-1198 | 62 |
| Guanaco 65B | 1065 | 974-1153 | 56 |
| ChatGPT 3.5 | 1023 | 936-1108 | 50 |
| Vicuna 13B | 1001 | 911-1091 | 47 |
| LLaMA 30B RLHF | 925 | 835-1017 | 36 |
| LLaMA 30B SFT | 935 | 843-1028 | 37 |
| text-davinci-003 | 800 | 712-893 | 22 |
| Human Evaluation | | | |
| STEERLM 43B | **1040** | 951-1126 | **59** |
| ChatGPT 3.5 | 981 | 897-1070 | 50 |
| Guanaco 65B | 977 | 890-1064 | 50 |

Table 2: Elo Ratings for Models based on Automatic and Human Evaluation.

Based on Tables 1 and 2, our STEERLM 43B model out-performs all baseline models on both automatic and human evaluations. STEERLM 43B performs slightly better than STEERLM 13B, which is expected given the larger base model size (Ouyang et al., 2022; Touvron et al., 2023). Analysis of the responses generated by STEERLM 43B shows that it provides longer, more complete answers (mean = 1906 characters) with more unique words (mean = 144) compared to the other baselines (e.g. ChatGPT 3.5 has 1193 characters and 77 unique words). Please refer to Appendix §A.6 for examples and §A.7 for average response lengths for each model.

Automatic evaluation with GPT-4 has a tendency to prefer longer responses that have more unique tokens (Dubois et al., 2023; Wang et al., 2023a). STEERLM 43B responses satisfy both criteria and this might lead to GPT-4 rating its responses higher

| Model Name | ChatGPT Score | Model Score | % of ChatGPT |
|---|---|---|---|
| Baseline | 663.50 | 528.50 | 79.7 |
| + Human Annotated Attributes | 644.00 | 619.50 | 96.2 |
| + HQ OASST (human annotated) | 646.75 | 634.75 | 98.1 |
| + Attribute Pred. Model | 632.50 | 649.75 | 102.7 |
| + HH-RLHF/M-SID | 623.25 | 646.00 | 103.7 |
| + Bootstrapping on HQ samples | 629.25 | 655.75 | **104.2** |

Table 3: Ablation study of model variants under automatic evaluation.

than other counterparts, explaining the 74 Elo point difference between STEERLM 43B and the closest baseline model on Automatic Evaluations. The advantage of STEERLM 43B on Human Evaluation is slightly lower (59 Elo point difference), suggesting that only a slight bias of GPT4 Automatic Evaluations towards longer and more informative responses. Nonetheless, both Automatic and Human Evaluation show a clear strength of STEERLM 43B in answering open-ended questions that require creative and verbose responses, which composes majority of the prompts in the Vicuna benchmark.

Relative to Guanaco 65B, our model performs better despite being trained on a smaller base model (43B vs. 65B) that has been trained on fewer tokens (1.1T vs. 1.4T). This is likely due to a more efficient use of OASST data, which both models are trained on. Briefly, Guanaco 65B only uses high quality data, defined as the top response (in terms of annotated quality) at every level of the conversation tree. In contrast, STEERLM 43B uses all of Open-Assistant data, including low-quality conversations. We will show the advantage of the STEERLM approach in the ablation studies in §5. This enables STEERLM to achieve an effect similar to RLHF by being able to learn from both responses of high and low quality, as opposed to only selected high-quality conversations in regular SFT (Dettmers et al., 2023; Wang et al., 2023a; Zhou et al., 2023; Peng et al., 2023; Köpf et al., 2023b; Chiang et al., 2023). The small difference in performance between the OASST LLaMA 30B SFT and RLHF models supports the noted difficulty of getting RLHF to work well in aligning LLMs (Ouyang et al., 2022; Bai et al., 2022; Köpf et al., 2023b), which prevents widespread adoption of RLHF.

## 5 Ablation Study

In order to identify the contribution of each individual component to the overall performance, we perform a systematic ablation process (Table 3),

starting from a baseline that is finetuned on the entire OASST dataset without using any information about attributes.

**Addition of attribute labels** The addition of attribute labels in the fine-tuning process leads to a significant increase in performance, underscoring the pivotal role of attribute labels, particularly the *quality* attribute, as the primary contributor to improved performance (16.5%).

**Finetuning on only High Quality Data** We observe that Attribute Conditioned Supervised Fine-tuning on a small subset of 3400 samples from the OASST dataset that have the highest value for *quality* (human annotated) leads to an increase in performance by up to 1.9%. This observation aligns with prior research indicating that training models on a larger volume of low-quality data yields inferior results compared to training on a smaller quantity of high-quality data (Dettmers et al., 2023; Zhou et al., 2023).

**Utilizing predictions from the Attribute Prediction model** for the Attribute Conditioned SFT process provides a substantial benefit to STEERLM 43B amounting to 4.6% in performance, relative to using human annotations. This empirical evidence substantiates our hypothesis that employing an attribute prediction model aids in effectively calibrating the quality annotation process across multiple training samples, thereby mitigating noisy labels that are unavoidable when using crowdsourced annotations.

**Augmentation of training data with Anthropic HH-RLHF** contributes to an additional 1.0% increase in performance compared to using Open-Assistant data alone. This shows the capability of our Attribute Prediction model to identify and annotate high-quality data from other datasets apart from the one it was trained on. This suggests that the methodology of STEERLM can be applied to label different kinds of datasets, enhancing the diversity of training data without incurring the substantial expense associated with human annotation for every individual data sample.

**Bootstrapping with High-Quality Samples** results in a performance gain of 0.5%. Our hypothesis is that the improvement in performance is due to the data augmentation through sampling, which explores the response space and augments the dataset with higher-quality data.

| Model Name | Toxicity Value | Avg. Score |
|---|---|---|
| STEERLM 43B | 0 | **0.06346** |
| | 3 | 0.08635 |
| | 6 | 0.09633 |
| | 9 | 0.10408 |
| ChatGPT 3.5 | - | 0.06622 |

Table 4: Average Perspective API Toxicity score on the Anthropic Red Team Dataset.

## 6 Steerability demonstration

To demonstrate the efficacy of STEERLM 43B to control its generations in a multi-faceted way, we conduct an empirical study incorporating the attributes of *toxicity* and *humor*.

### 6.1 Toxicity

To assess the ability of STEERLM 43B to vary its responses based on the value of toxicity specified, we use the Anthropic Red-team dataset[3] (Ganguli et al., 2022). This dataset contains prompts ranked from 0 to 4, with a higher rank indicating more inappropriate responses. We randomly select 100 conversations with a ranking of 4 and focused on the initial turns of these interactions.

To compare the toxicity levels of generated outputs, we vary the toxicity parameter in STEERLM 43B at four different settings: 0 (default), 3, 6, and 9 (highest toxicity). We employ the Perspective API, a widely-used toxicity classification model, to score the toxicity of the generated responses and compare the average toxicity of STEERLM 43B to the responses from ChatGPT-3.5.

Our findings (Table 4) indicate that when the toxicity value is set to 0 (default), STEERLM 43B exhibits slightly lower toxicity compared to ChatGPT. However, STEERLM 43B offers the flexibility to generate more toxic outputs by increasing the toxicity value above 0. This ability to control generation attributes at inference can prove valuable in applications such as generating Non-Player Character (NPC) dialogue in games and red-teaming purposes.

### 6.2 Humor

Recent studies (Jentzsch and Kersting, 2023) investigating the humor capabilities of language models have primarily focused on the aspect of telling jokes. However, humor can manifest in various

---

[3]Accessed at https://huggingface.co/datasets/Anthropic/hh-rlhf/tree/main/red-team-attempts

contexts. In this experiment, we employ two different prompts - *"Tell me a joke"* and *"What's a good way to spend a day?"*, to compare the humor exhibited by STEERLM and ChatGPT-3.5.

**"Tell me a joke"** - ChatGPT-3.5 and STEERLM 43B, with humor set to 9, successfully generate a joke. When the humor attribute in STEERLM 43B is set to 6 or lower, it responds with *"I don't know any jokes"*. This characteristic can be advantageous for chatbots that need to maintain a formal persona and when humor is not appropriate.

**"What's a good way to spend a day?"** - Both ChatGPT-3.5 and STEERLM 43B, with humor attribute at low values (0-3) provided an extensive list of activities. Tuning up humor to 6 leads to STEERLM 43B appending to the list a cheeky remark, *"But don't forget to brush your teeth and go to bed at a reasonable hour!"*. With humor set to the maximum value of 9, it leads to a cheesy statement "By spending it with you, of course!".

Thus, depending on the intended use-case the same STEERLM model can be turned into versatile engaging conversational agents.

## 7   Conclusion

We introduce STEERLM, a novel model alignment approach with a value system (e.g. humor level and toxicity tolerance) that can be adjusted by users at inference time without re-training. STEERLM trains both the attribute prediction model and the language model using only supervised fine-tuning, resulting in an easy-to-implement and straightforward training process compared to using RLHF. We train STEERLM models following this procedure, achieving state-of-the-art results on the Vicuna benchmark. We validate these results with a human evaluation and find that STEERLM is preferred over the other models we compare it to. We hope our work will inspire further research into developing simple and effective model alignment methods that empower better AI assistants for everyone.

## Limitations

The Attribute Prediction Model and Attribute-Conditioned SFT Models in STEERLM are fully supervised fine-tuned, making it relatively costly in terms of GPU hours and energy compared with Parameter Efficient Fine-Tuning algorithms such as Low-Rank Adaptation and Prompt Tuning techniques.

Automatic and manual evaluation have been performed on a English-only benchmark and we would need to evaluate our models on multilingual benchmarks to test the extent to which the positive results we find with STEERLM on an English-based benchmark extends to evaluations with other languages.

## Ethics Statement

STEERLM enables user capabilities to ask for toxic and/or violent responses during run-time. This is a valid use case in certain scenarios like games where a character should respond differently based on situation or in red teaming use cases. However, malicious users might exploit this feature outside of its intended use-cases. To address this, it is possible to give users controls over only a subset of attributes at model run-time, unless they demonstrate an acceptable use-case. In general, we believe that giving developers explicit control over model's value system is preferred to dictating one that they can not adjust.

## Acknowledgments

We would like to thank Abhinav Khattar, Adi Renduchintala, Neel Kant, Sandeep Subramanian and many others at NVIDIA as well as anonymous reviewers for their helpful comments. In addition, we appreciate the authors, members and contributors of the Open Assistant Project (OASST) for assembling and excellent open data set for model alignment which made this work possible.

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

# A  Appendix

## A.1  Attribute Prediction Model prompt templates

We use the following prompt template to train the value model. The encoded value string is marked in red and used to calculate LM loss. The <extra_id_0>,<extra_id_1> and <extra_id_2> are special tokens included in the base LM tokenizer. An example of linearized value attributes is shown in the Sec. 3.1.

```
<extra_id_0>System
[system prompts]
<extra_id_0>[user name]
[user prompts]
<extra_id_1>[assistant name]
[assistant response]
<extra_id_2>[encoded value attributes]
<extra_id_0>[user name]
[user prompts]
<extra_id_1>[assistant name]
[assistant response]
<extra_id_2>[encoded value attributes]
```

## A.2  Attribute Conditioned SFT prompt templates

The SteerLM ACSFT prompt templates is similar to the Attribute Prediction Model prompt template shown in the Sec. A.1 but we swap the position of assistance response and encoded value string. The LM loss is calculated on the assistant responses as highlighted in red.

```
<extra_id_0>System
[system prompts]
<extra_id_0>[user name]
[user prompts]
<extra_id_1>[assistant name]
<extra_id_2>[encoded value attributes]
[assistant response]
<extra_id_0>[user name]
[user prompts]
<extra_id_1>[assistant name]
<extra_id_2>[encoded value attributes]
[assistant response]
```

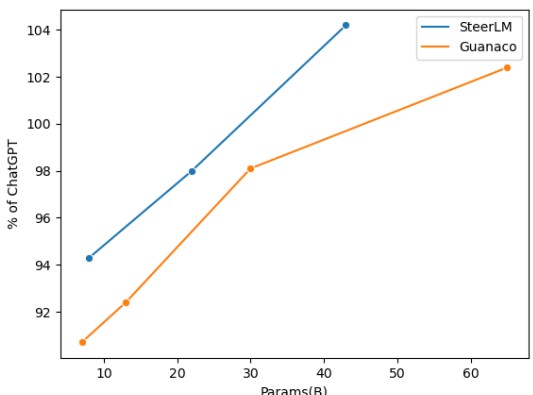

Figure 3: Effect of Model Size

## A.3  Further Data Collection

While we did not collect new data in the current study, users who would like to collect new data will find the annotation cost of SteerLM to substantially lower compared to RLHF. SteerLM only requires each prompt-response sample to be annotated with attributes of their interest whereas RLHF requires pairwise comparison between k responses per prompt (where k=4-9) or effectively $\binom{4}{2} = 6$ to $\binom{9}{2} = 36$ annotation per prompt (Ouyang et al., 2022). RLHF also relies on an iterative data collection methodology, where the critic must be continually retrained as the actor (language model) improves (Ouyang et al., 2022; Bai et al., 2022; Touvron et al., 2023). In contrast, SteerLM only requires a one-time data collection, as each response is rated independently on an absolute scale rather than compared to other responses. Therefore, the annotation cost of collecting a dataset for SteerLM is approximately an order of magnitude lower than RLHF.

## A.4  Effect of model size on performance

To better understand the effect that model size plays in model performance, we also trained SteerLM on smaller base models (8B and 22B), which have pretrained similarly as the 43B base model. We then compare it to smaller versions (7B , 13B and 30B) of the best performing baseline model (Guanaco 65B). As shown in Figure 3, SteerLM outperforms Guanaco at every comparable model size.

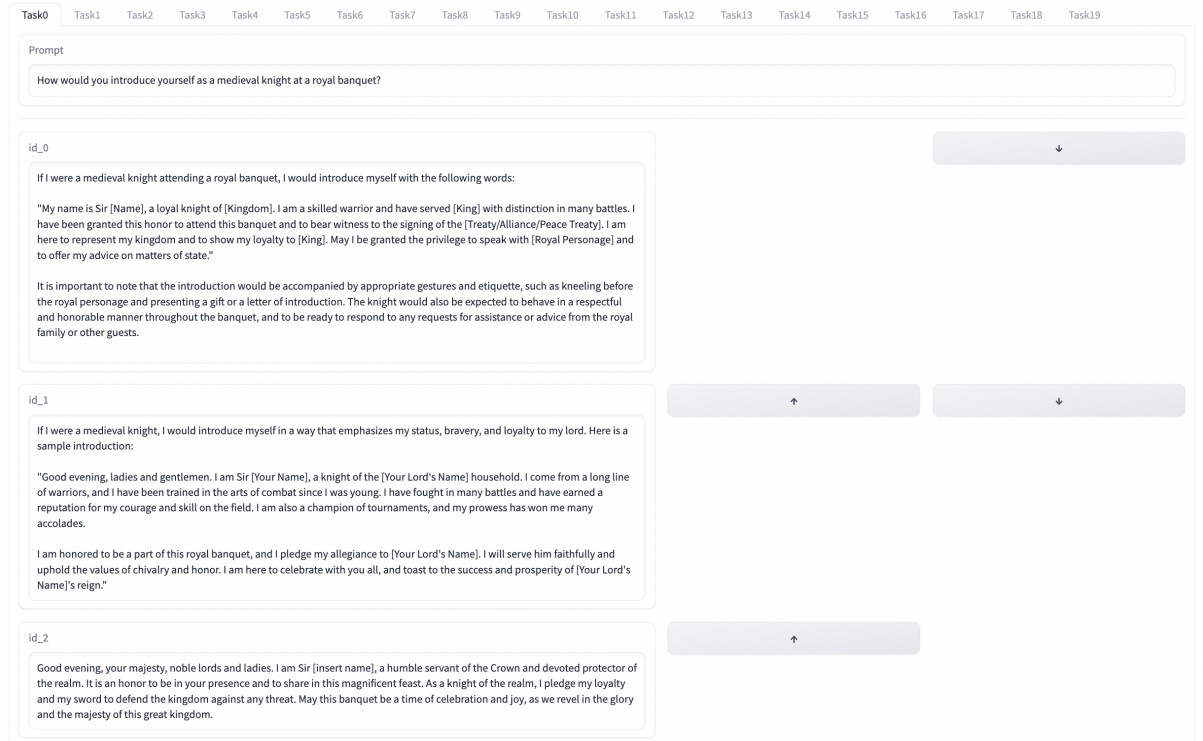

Figure 4: Human Annotation UI

## A.5 Human Annotation UI

## A.6 Example of Generations by SteerLM 43B and ChatGPT 3.5

**Question:** How can I increase my productivity while working from home?

**SteerLM 43B response:** Here are some tips to increase your productivity while working from home:

1. Create a dedicated workspace: Having a dedicated workspace can help you focus and avoid distractions. Choose a quiet and comfortable spot in your home and set up your workspace with all the necessary tools and equipment.

2. Stick to a schedule: Establish a daily routine and stick to it. Set specific hours for work and stick to them. Take breaks at regular intervals and schedule time for lunch and other activities.

3. Eliminate distractions: Minimize distractions by turning off your phone, closing unnecessary tabs on your computer, and letting your family and friends know that you are working and cannot be disturbed.

4. Take care of your physical health: Take regular breaks to stretch and move around. Exercise regularly and eat healthy foods to keep your energy levels up.

5. Stay connected: Communicate with your colleagues and clients regularly. Use video calls and instant messaging to stay in touch and avoid misunderstandings.

6. Set clear goals: Set specific and achievable goals for each day or week. Break down large tasks into smaller ones and prioritize your tasks based on importance and urgency.

7. Use productivity tools: There are many productivity tools available that can help you stay organized and focused. Use tools like calendars, to-do lists, and project management software to keep track of your work.

8. Take breaks: Taking regular breaks can help you stay focused and productive. Take a short break every hour or so to stretch your legs, get some fresh air, or grab a snack.

9. Stay motivated: Find ways to stay motivated and inspired. Set short-term and long-term goals, celebrate your successes, and find ways to reward yourself for a job well done.

10. Be flexible: Working from home can be challenging, so be flexible and adapt to changes as they arise. Be open to new ideas and approaches, and don't be afraid to ask for help when you need it.

**ChatGPT 3.5 response:** There are several ways to increase productivity while working from home:

1. Establish a routine: Start and end your work at a specific time each day, just like you would if you were working at an office.

2. Create a designated workspace: Set up a workspace that is free from distractions and has everything you need to complete your work.

3. Prioritize your tasks: Make a to-do list at the beginning of each day and prioritize your tasks based on their importance.

4. Take breaks: Take regular breaks to rest your mind and stretch your body, as this can help improve your focus and productivity.

5. Minimize distractions: Avoid distractions such as social media, TV, or household chores during your work hours.

6. Use productivity tools: There are several productivity tools available online, such as time trackers, task-management apps, and project-management software, which can help increase your productivity.

7. Communicate with colleagues: Keep in touch with your colleagues regularly to stay connected and avoid feeling isolated.

Remember, everyone's productivity habits are different, so find what works best for you and stick to it.

## A.7 Response lengths of various models

| Model | Characters | Unique Words |
|---|---|---|
| STEERLM 43B | 1906 | 144 |
| STEERLM 13B | 1719 | 146 |
| Guanaco 65B | 1648 | 145 |
| ChatGPT 3.5 | 1193 | 77 |
| Vicuna 13B | 1417 | 125 |
| LLaMA 30B RLHF | 1219 | 108 |
| LLaMA 30B SFT | 1280 | 106 |
| text-davinci-003 | 842 | 87 |

Table 5: Mean response length for each model in characters and number of unique whitespace-separated words.