# OpenReview forum: "SteerLM: Attribute Conditioned SFT as an (User-Steerable) Alternative to RLHF"
_EMNLP/2023/Conference — EMNLP 2023 Findings_

### Official Review · Reviewer_QL8N · 2023-08-04

**Soundness:** 4

**Excitement:**

4: Strong: This paper deepens the understanding of some phenomenon or lowers the barriers to an existing research direction.

**Paper Topic And Main Contributions:**

Propose StterLF which first trained an attribute predict model and use it filter high quality data for retraining.
It allows user to decode with its user preference based on attributes.
It evaluates on Vicuna test set and performs better than ChatGPT 3.5 in auto evaluation and human evaluation.

**Questions For The Authors:**

(A) This paper shows superior performance than ChatGPT which is too good to be true. Please provide more details on your evaluation including does it contain math and code related tasks and please report the results on these difficult tasks since we know open-sourced chat models like Alpaca, Vicuna all perform bad on math and code tasks.
(B) Automatic evaluator like GPT-4 may give a higher score for very long responses, you should provide an average response length for reponses.

**Reasons To Accept:**

Retraining using bootstrap high-quality data is sound and it makes user can decide the attributes like humor or toxicity
It outperforms ChatGPT 3.5 in auto evaluation and human evaluation on Vicuna test.
Detailed ablation studies proved its effective design.

**Reasons To Reject:**

See questions.

**Reproducibility:**

2: Would be hard pressed to reproduce the results. The contribution depends on data that are simply not available outside the author's institution or consortium; not enough details are provided.

**Reviewer Confidence:**

5: Positive that my evaluation is correct. I read the paper very carefully and I am very familiar with related work.

---

> ### Author Rebuttal · Authors · 2023-08-28
>
> Dear Reviewer,
>
> Thank you for the positive feedback and for raising important questions to help us strengthen the paper. Please find our responses below:
>
>
> >This paper shows superior performance than ChatGPT which is too good to be true. Please provide more details on your evaluation including does it contain math and code related tasks and please report the results on these difficult tasks since we know open-sourced chat models like Alpaca, Vicuna all perform bad on math and code tasks.
>
> In the current version of SteerLM, we observe improved performance on open-ended generation tasks such as roleplay, creative writing and brainstorming, which characterizes most of Vicuna Bench. While our paper follows standard practice in only sharing the model’s overall performance on Vicuna Benchmark (Chang et al., 2023 [1], Dettmers et al., 2023 [2]), we agree that it is important to analyze performance on these non-textual tasks. We find in our internal experiments that our base 43B model is inferior to ChatGPT-3.5 on the math and coding benchmarks which is not reflected in the automatic eval using GPT-4 on the Vicuna benchmark.
>
> The weakness in the coding and math categories is common among the open source language models including the recent LLaMa 2 model. Code Llama (Rozière, 2023[5]) shows that this can be fixed by continual training on the massive amount of code data, which is outside the scope of typical model alignment methods.  We are working on improving the math and coding performance of our base LMs and we hope to include some improved results in the camera ready version.
>
>
>
>
>
>
> >Automatic evaluator like GPT-4 may give a higher score for very long responses, you should provide an average response length for responses.
>
>
>
>
> We appreciate you raising this observation, which echoes findings in literature (Dubois et al., 2023[3] ; Wang et al., 2023 [4]).  Below are the average length of responses from each model, which we will include in the camera ready version.
>
>
> | Model            | Avg. Response Chars |
> |------------------|---------------------|
> | SteerLM 43B      | 1906                |
> | Guanaco 65B      | 1648                |
> | Vicuna 13B       | 1417                |
> | Llama 30B SFT    | 1280                |
> | Llama 30B RLHF   | 1219                |
> | Text-davinci-003 | 842                 |
> | ChatGPT 3.5      | 1193                |
>
>
>
>
> Models with longer responses do perform better on Vicuna Bench, according to the GPT4 automatic evaluator. To make sure this bias towards longer responses is reflective of human preferences, we also performed Human Evaluation of model responses and found that human annotators prefer the longer responses generated by  SteerLM 43B compared to those by Guanaco 65B and ChatGPT 3.5.
>
>
>
>
> >Reproducibility
>
>
> At the time of conducting our experiments prior to EMNLP submission, we could not access publicly available models with competitive performance, since they are only licensed to non-commercial settings,. Therefore, we had to develop our SteerLM recipe using an in-house model.
>
> However, since then, models with licenses permitting use by commercial labs (e.g. Llama 2) have been released. Therefore, we are working towards using our SteerLM recipe on publicly accessible models prior to releasing the camera-ready version of our paper. As part of that effort, we have trained a SteerLM Llama2 13B model which reports 102.6% of ChatGPT 3.5 performance on Vicuna Bench. We will also release code for training various base models using SteerLM after paper acceptance. Together, we hope this can alleviate some of your concerns regarding the reproducibility of our approach.
>
>
> >Conclusion
>
>
> Thank you again for the constructive feedback - addressing your comments will significantly strengthen the rigor and transparency of our paper. We look forward to submitting an improved camera ready version.
>
> >References
>
> [1] Chiang et al. (2023) Vicuna: An open-source chatbot impressing gpt-4 with 90%* chatgpt quality.
>
> [2] Dettmers et al. (2023) QLoRA: Efficient Finetuning of Quantized LLMs
>
> [3] Dubois et al., (2023) Alpaca-farm: A simulation framework for methods that learn from human feedback
>
> [4] Wang et al., (2023) How far can camels go? exploring the state of instruction tuning on open resources.
>
> [5] Rozière et al., (2023) Code Llama: Open Foundation Models for Code

---

### Official Review · Reviewer_PUsH · 2023-08-05

**Soundness:** 3

**Excitement:**

3: Ambivalent: It has merits (e.g., it reports state-of-the-art results, the idea is nice), but there are key weaknesses (e.g., it describes incremental work), and it can significantly benefit from another round of revision. However, I won't object to accepting it if my co-reviewers champion it.

**Missing References:**

[1] RAFT: Reward rAnked FineTuning for Generative Foundation Model Alignment https://arxiv.org/abs/2304.06767

**Paper Topic And Main Contributions:**

This paper introduces SteerLM, a supervised fine-tuning method that allows users to control the responses generated by language models. The authors address the limitations of reinforcement learning from human feedback (RLHF) by proposing a method that conditions responses to conform to a set of explicitly defined attributes. They demonstrate that SteerLM outperforms state-of-the-art baselines trained with RLHF on the Vicuna benchmark, while being easier to train. The paper provides detailed explanations of the methodology and presents results from automatic and human evaluations.

**Reasons To Accept:**

(1) The paper proposes a novel approach, SteerLM, which offers a simpler and more user-controlled alternative to RLHF for aligning language models with human preferences.

(2) The methodology is well-structured and easy to understand. The authors provide clear explanations of the attribute prediction model, attribute-conditioned SFT, and bootstrapping with high-quality samples.

(3) The paper includes comprehensive experimental evaluations, including automatic evaluation with GPT-4, human evaluation, and an ablation study, to demonstrate the effectiveness of STEER LM.

**Reasons To Reject:**

(1) More details are needed for the base model. What is the base performance? How much compute does it costs? Are there any public access? Because it is unclear about the base model, we do not know how much improvement has been made by the proposed method.

(2) There are 13 attributes in OASST, why do the authors choose 7 out of 13?

(3) The method is not new. It is needed to discuss the differentiation on the method with RAFT [1] and RRHF [2]. Especially, an important and similar work is omitted [1].

[1] RAFT: Reward rAnked FineTuning for Generative Foundation Model Alignment https://arxiv.org/abs/2304.06767

[2] Rrhf: Rank responses to align language models with human feed- back without tears https://arxiv.org/abs/2304.05302

**Reproducibility:**

3: Could reproduce the results with some difficulty. The settings of parameters are underspecified or subjectively determined; the training/evaluation data are not widely available.

**Reviewer Confidence:**

4: Quite sure. I tried to check the important points carefully. It's unlikely, though conceivable, that I missed something that should affect my ratings.

---

> ### Author Rebuttal · Authors · 2023-08-28
>
> Dear Reviewer,
>
> Thank you for the thoughtful feedback on our paper submission. We appreciate you identifying areas where we can provide additional details to strengthen the work. Please find our responses below:
>
>
> >More details are needed for the base model. What is the base performance? How much compute does it costs? Are there any public access? Because it is unclear about the base model, we do not know how much improvement has been made by the proposed method.
>
> We agree that including more details about the base model can help to better clarify the contributions of SteerLM methodology. At the time of conducting our experiments prior to EMNLP submission, we could not access publicly available models with competitive performance, since they are only licensed to non-commercial settings. Therefore, we had to develop our SteerLM recipe using an in-house model.
>
> Here are details of our in-house 43B model:
>
> |             | In House 43B | Llama 65B (used by Guanaco 65B) |
> |-----------|--------------|---------------------------------|
> | MMLU      | 55.3         | 63.4                            |
> | ARC-Easy  | 76.6         | 78.9                            |
> | HellaSwag | 80.1         | 84.2                            |
>
>
> Compute: $2.9 * 10^{23}$ FLOPs
>
> Public access: Currently, 43B model is not publicly accessible and can only be accessed by our enterprise users.
>
> To better understand the contributions of the SteerLM methodology to the performance of our model, we also trained a 43B model on the Guanaco dataset used by Guanaco 65B. Guanaco 43B performs at 98.4% of ChatGPT 3.5, which is much lower than 102.4% achieved by Guanaco 65B (with Llama 65B base model). This suggests that training on the inhouse 43B model puts us at a disadvantage compared to training on Llama 65B, as also supported by the lower performance of our base model on MMLU, ARC-Easy and HellaSwag. In other words, the SteerLM approach contributes to the good performance on the Vicuna Bench, in spite of a weaker 43B base model used.
>
> Since our EMNLP submission, models with licenses permitting use by commercial labs (e.g. Llama 2) have been released. Therefore, we are working towards using our SteerLM recipe on publicly accessible models prior to releasing the camera-ready version of our paper. As part of that effort, we have trained a SteerLM Llama2 13B model which reports 102.6% of ChatGPT 3.5 performance on Vicuna Bench. This performance is not far off the SteerLM 43B (104.2%), which is reasonable given the model size disparities. This supports how SteerLM approach can be used to align various base models, independent of the contribution of the base 43B model.
>
>
>
> >There are 13 attributes in OASST, why do the authors choose 7 out of 13?
>
>
> We chose 7 attributes based on their relevance for steering the model's responses during inference to match human preferences. Other attributes such as hate_speech, lang_mismatch and pii (personal identifiable information) are not useful to steer at inference time since these are attributes that we always want to keep as False. We will take care to make this decision clearer in our paper.
>
>
>
>
>
>
> >The method is not new. It is needed to discuss the differentiation on the method with RAFT [1] and RRHF [2]. Especially, an important and similar work is omitted [1].
> [1] RAFT: Reward rAnked FineTuning for Generative Foundation Model Alignment https://arxiv.org/abs/2304.06767 [2] Rrhf: Rank responses to align language models with human feed- back without tears https://arxiv.org/abs/2304.05302
>
>
>
> Thank you for bringing these papers to our attention. After better understanding these papers, we found areas of differences that make SteerLM distinct from RAFT and RRHF. We discuss these differences below and will include this discussion in our camera-ready version.
>
>
> SteerLM conditions generation on explicit, multi-dimensional attributes that can be specified by the user at inference time. This enables user-steerable responses, which can be useful in settings such as generating game dialogues, in which some NPCs have toxic/unhelpful/humorous persona.  In contrast, RAFT and RRHF optimize a scalar reward signal without explicit conditioning on attributes, limiting its user-controllability at inference time.
>
> SteerLM trains the model to generate responses of both high and low scores in each attribute in steps 1 to 3, meaning that models learn to generate responses conditioned on both high and low attribute scores. This is important for steerability because users might want responses either high or low on humor and other attributes.. On the other hand, RaFT seeks to train models using only samples with high reward scores (Algorithm 1 Step 2) and RRHF seeks to optimize $L_{rank}$ (Eq 2) and $L_{ft}$ (Eq. 4) to maximize the likelihood of generating responses with high rewards. Therefore, the trained RaFT and RRHF models can only generate responses with high rewards but not low rewards.
>
> SteerLM is evaluated on Vicuna Benchmark, which covers a diverse set of prompts from various domains and shows SOTA performance with 104.2% of ChatGPT 3.5 performance. RRHF only reaches 81.9% of ChatGPT performance on Vicuna Benchmark (Table 9) while RAFT is only evaluated on tasks in narrow domains (specifically movie review generation, dialogue-completion and harmless/helpfulness classification).
>
>
> However,  we note that both RAFT and RRHF papers were submitted to Arxiv in April 2023 and are within the contemporaneous 3 month period from the EMNLP submission deadline (23 June 2023). Therefore, they should not constitute an acceptable reason for rejection, based on guidance from EMNLP.
>
>
> ```While not citing such unpublished works upon submission is not sufficient grounds for paper rejection, you are expected to cite such relevant work in camera ready, if notified about it by reviewers.```
>
>
> We will make sure to cite these works and include the discussion. We request the reviewer to increase the score since the EMNLP guidelines state that this is not sufficient grounds for rejection.
>
>
>
> >Conclusion
>
> Thank you again for the helpful feedback. We hope that we have clarified your concerns and therefore request you to favorably reconsider the scores for our paper.

---

### Official Review · Reviewer_kdwP · 2023-08-05

**Soundness:** 3

**Excitement:**

3: Ambivalent: It has merits (e.g., it reports state-of-the-art results, the idea is nice), but there are key weaknesses (e.g., it describes incremental work), and it can significantly benefit from another round of revision. However, I won't object to accepting it if my co-reviewers champion it.

**Paper Topic And Main Contributions:**

The paper introduces SteerLM, a novel approach for model alignment that allows users to adjust the value system (e.g., humor level and toxicity tolerance) during inference without requiring re-training. The authors utilize a specified supervised fine-tuning (SFT) method for implementing alignment. They trained a 43B model using SteerLM instead of RLHF and achieved state-of-the-art results on the Vicuna benchmark.

**Questions For The Authors:**

1. How does the data collection cost for attribute conditioned SFT compare to RLHF? It is important to consider the annotation costs associated with RLHF.
2. Regarding the SteerLM pretrained model, how can we determine the relative contributions of the pretraining phase and the proposed attribute conditioned SFT?
3. Is it more appropriate to consider attribute conditioned SFT as an alternative to SFT, or does it serve a different purpose?
4. In addition to the Vicuna evaluation, could the paper include more evaluation results specifically focusing on toxicity and hallucinations?
5. Providing more examples related to toxicity and hallucinations would enhance the understanding of SteerLM's performance in these areas.

**Reasons To Accept:**

1. The paper proposes an alternative method for model alignment, providing a different approach to RLHF.
2. The proposed SteerLM approach demonstrates strong performance, surpassing existing models on the Vicuna benchmark.

**Reasons To Reject:**

1. The proposed method may introduce challenges in collecting alignment data, potentially increasing the difficulty of the alignment process.
2. It is important to verify the proposed method on other benchmarks or approaches to determine its generalizability. It remains unclear whether the superior performance is primarily attributed to the 43B pretrained model or the attribute conditioned SFT.
3. The paper would benefit from additional evaluation results. Existing benchmarks lack evaluations specifically related to toxicity and hallucinations, which are crucial aspects for assessing alignment effectiveness.

**Reproducibility:**

3: Could reproduce the results with some difficulty. The settings of parameters are underspecified or subjectively determined; the training/evaluation data are not widely available.

**Reviewer Confidence:**

3: Pretty sure, but there's a chance I missed something. Although I have a good feel for this area in general, I did not carefully check the paper's details, e.g., the math, experimental design, or novelty.

---

> ### Author Rebuttal · Authors · 2023-08-28
>
> Dear Reviewer,
>
> Thank you for your thoughtful feedback on our paper. We appreciate you taking the time to provide constructive criticism to help strengthen our work. Please find our responses below:
>
>
>
> >The proposed method may introduce challenges in collecting alignment data, potentially increasing the difficulty of the alignment process.
> How does the data collection cost for attribute conditioned SFT compare to RLHF? It is important to consider the annotation costs associated with RLHF.
>
> We appreciate the reviewer highlighting the importance of considering annotation costs when comparing alignment approaches. SteerLM does not require any additional annotations since we only use existing annotations from the open-sourced Open Assistant dataset. Taking a step further, if other researchers wish to apply the SteerLM methodology by training on another dataset, the annotation cost of SteerLM will be drastically lower compared to RLHF.
>
> This is because SteerLM only requires each prompt-response sample to be annotated with attributes of their interest (can be as few as 1 - for overall quality) whereas RLHF requires pairwise comparison between k responses per prompt (where k=4 to 9) or effectively $ 4 \choose 2 $ to $ 9 \choose 2 $ = 6 to 36 annotation per prompt (Ouyang et al., 2022 [1]). Therefore, the annotation cost of collecting a dataset for SteerLM is approximately an order of magnitude lower than RLHF.
>
> RLHF also relies on an iterative data collection methodology, where the critic must be continually retrained as the actor (language model) improves. This is because the actor can easily fool the outdated critic, requiring new human preference rankings for the updated actor in order to train an improved critic (Ouyang et al., 2022 [1]). In contrast, SteerLM only requires a one-time data collection, as each response is rated independently on an absolute scale rather than compared to other responses. So while RLHF needs repeated data collection and critic retraining as the actor evolves, SteerLM provides a more efficient training approach with a single round of human ratings that do not need to be updated as the model improves.
>
>
> We will add these discussions in the camera ready version to further clarify the significant annotation cost advantages of our approach compared to RLHF.
>
>
>
>
> >It remains unclear whether the superior performance is primarily attributed to the 43B pretrained model or the attribute conditioned SFT.
> Regarding the SteerLM pretrained model, how can we determine the relative contributions of the pretraining phase and the proposed attribute conditioned SFT?
>
>
> We acknowledge that our original submission does not sufficiently clarify the impact of the base model vs the alignment methodology. To address this concern, we have conducted some additional experiments which provide evidence that the performance improvement comes from the proposed SteerLM methodology rather than from having a better pretrained model.
>
> Specifically, we evaluated our 43B model by SFTing on the Guanaco dataset and it achieved a performance of 98.4% of ChatGPT-3.5. On the other hand, training LLaMa 65B on the same dataset achieved a performance of 102.4% of ChatGPT (Dettmers et al., 2023 [2]). This demonstrates that the superior performance cannot be attributed to the base model, and should instead be attributed to the SteerLM methodology.
>
> At the time of conducting the experiments before the submission, we could not access and use the publicly available models which are only licensable to non-commercial settings. However, since then, there have been open-source models that have been released. In order to also show the generalizability of our proposed methodology, we use LLaMa 2 - 13B as our base model and apply our SteerLM methodology. Evaluating it on the Vicuna benchmark, we find that it gets a score that is 102.6% of ChatGPT-3.5 score, which is not far off the SteerLM 43B (104.2%), a reasonable difference given the model size disparities. This further supports that the superior performance can be attributed to the SteerLM methodology.
>
> In the camera ready version, we will include these new experimental results and analysis to better isolate the contributions of our approach.
>
>
>
> >Is it more appropriate to consider attribute conditioned SFT as an alternative to SFT, or does it serve a different purpose?
>
> We position attribute conditioned SFT as an augmentation of regular SFT that enables incorporating reward signal information through attribute labels. This allows learning from both high and low quality responses in a manner similar to the established SFT+RLHF pipeline (Ouyang et al., 2022 [1]). In the camera-ready version, we will provide further clarification on the similarities and differences between our proposed approach and regular SFT.
>
>
>
>
> >It is important to verify the proposed method on other benchmarks or approaches to determine its generalizability.
> The paper would benefit from additional evaluation results. Existing benchmarks lack evaluations specifically related to toxicity and hallucinations, which are crucial aspects for assessing alignment effectiveness.
> In addition to the Vicuna evaluation, could the paper include more evaluation results specifically focusing on toxicity and hallucinations? Providing more examples related to toxicity and hallucinations would enhance the understanding of SteerLM's performance in these areas.
>
>
>
>
> We have included in section 6.1 a brief case study on how we evaluated SteerLM on the toxicity of its generations when the attribute of toxicity is explicitly controlled for. Additionally, we agree with the reviewer on the need for further evaluations targeting hallucinations. However, there has been a lack of well-formulated, stable benchmarks that evaluate for such criteria as pointed out by the reviewer themselves. Thus we would like to state that this is not an appropriate reason for rejection for the contributions of our work.  The main contribution of our work is a simple, easy-to-use method for aligning language models. While the claim of this paper is not to propose a method that can significantly reduce toxicity and hallucination compared to existing approaches, we feel that it is easy to control such attributes using SteerLM and can be explored in future work.
>
>
> >Conclusion
>
>
> Thank you again for the helpful feedback, which strengthens our paper.  We request the reviewer to look more favorably on our paper and increase the scores since some of the reasons for rejection are open research problems in the field and are not directly related to our main contributions.
>
>
> >References
>
>
> [1] Ouyang et al. (2022) Training language models to follow instructions with human feedback
>
> [2] Dettmers et al. (2023) QLoRA: Efficient Finetuning of Quantized LLMs

---

### Meta-Review · Area_Chair_ARCV · 2023-09-22

**Recommendation:** 4

**Metareview:**

In this work, the authors introduce SteerLM -- a new technique for model alignment that allows for specific attributes to be controlled by the end-users during inference (unlike existing alignment techniques such as RLHF). Experiments show that using SteerLM yields responses that are preferred in both human and automatic evaluations, compared to RLHF, and the training cost is also lower than RLHF.

The reviewers have all rated this work as good or higher on soundness. With moving to a larger model (43B), reviewers asked for more clarity on how to disentangle the positive effect of more pretraining data vs. the proposed attributed-conditioned finetuning. The authors responded with more results on models, that were released post the EMNLP submission, with licenses permitting use by commercial labs. The reviewers also pointed to very recent and relevant references that the authors have promised to include in their revised version. A concern about reproducibility was also addressed with a promise to release the SteerLM recipe on publicly available models.

---

### Decision · Program_Chairs · 2023-10-07

**Decision:**

Accept-Findings

**Comment:**

In this work, the authors introduce SteerLM -- a new technique for model alignment that allows for specific attributes to be controlled by the end-users during inference (unlike existing alignment techniques such as RLHF). Experiments show that using SteerLM yields responses that are preferred in both human and automatic evaluations, compared to RLHF, and the training cost is also lower than RLHF.

The reviewers have all rated this work as good or higher on soundness. With moving to a larger model (43B), reviewers asked for more clarity on how to disentangle the positive effect of more pretraining data vs. the proposed attributed-conditioned finetuning. The authors responded with more results on models, that were released post the EMNLP submission, with licenses permitting use by commercial labs. The reviewers also pointed to very recent and relevant references that the authors have promised to include in their revised version. A concern about reproducibility was also addressed with a promise to release the SteerLM recipe on publicly available models.